# Beyond Pain Relief: Unveiling the Multifaceted Impact of Anti-CGRP/R mAbs on Comorbid Symptoms in Resistant Migraine Patients

**DOI:** 10.3390/biomedicines12030677

**Published:** 2024-03-18

**Authors:** Alessandra Della Vecchia, Ciro De Luca, Lucrezia Becattini, Letizia Curto, Elena Ferrari, Gabriele Siciliano, Sara Gori, Filippo Baldacci

**Affiliations:** 1Section of Psychiatry, Department of Clinical and Experimental Medicine, University of Pisa, Via Roma, 67, 56100 Pisa, Italy; alessandradellavecchia@gmail.com; 2Laboratory of Neuronal Network Morphology and Systems Biology, Department of Mental, Physical Health and Preventive Medicine, University of Campania “Luigi Vanvitelli”, 80138 Naples, Italy; 3Neurology Unit, Department of Clinical and Experimental Medicine, University of Pisa, 56126 Pisa, Italy; lu.becattini@gmail.com (L.B.); letiziacurto@hotmail.it (L.C.); elena.ferrari91@gmail.com (E.F.); gabriele.siciliano@unipi.it (G.S.); sara.gori@ao-pisa.toscana.it (S.G.); filippo.baldacci@unipi.it (F.B.)

**Keywords:** migraine disorders, comorbidity, depression, anxiety, fatigue, allodynia, calcitonin gene-related peptide

## Abstract

The study aimed to evaluate the effects of monoclonal antibodies (mAbs) acting on the calcitonin gene-related peptide (CGRP) or its receptor (anti-CGRP/R mAbs) on migraine comorbidities of depression, anxiety, and fatigue in patients resistant to traditional therapies. The issue addressed in this study is pivotal to unveiling the role of this neurotransmitter beyond pain processing. We conducted an open-label prospective study assessing comorbidities in patients with high frequency (HFEM) and chronic migraine (CM), medication overuse headache (MOH), and resistance to traditional prophylaxis. All patients were treated with anti-CGRP/R mAbs for 3 months. Seventy-seven patients were enrolled with either HFEM (21%) or CM (79%) with or without MOH (56% and 44%, respectively). We identified 21 non-responders (27%) and 56 responders (73%), defined on the reduction ≥50% of headache frequency. The two groups were highly homogeneous for the investigated comorbidities. Disease severity in terms of headache frequency, migraine-related disability, and affective comorbid symptoms was reduced in both groups with different thresholds; allodynia and fatigue were ameliorated only in responders. We found that anti-CGRP/R antibodies improved pain together with affection, fatigue, and sensory sensitization in a cohort of migraine patients resistant to traditional prophylaxis. Our results offer novel perspectives on the early efficacy of anti-CGRP/R mAbs in difficult-to-treat patients focusing on clinical features other than pain relief.

## 1. Introduction

Migraine ranks as the most prevalent, disabling, and long-term neurological disease [1,2]. Migraine is an evolutive disease, in which the clinical features can vary over a long time course, probably according to different pathophysiological mechanisms [3].

The prevalence of migraine in the general population is around 15%, and 8% of migraineurs suffer from chronic migraine (CM) [4,5]. CM is more burdensome than episodic migraine (EM) in terms of disability, quality of life, use of health resources, involvement of comorbidities, and drug resistance [5,6]. If both EM and CM are considered, the disease has a paramount financial burden on Western economies, with an estimate of USD 19.6 billion in the United States [7] and EUR 27 billion in the European Union [8] per year. However, the true socioeconomic burden comprises the intangible costs and the reduced productivity of the affected population [9]. Migraine affects females and males differently with a 3:1 ratio, and impairs their quality of life during the most productive period of their lives [10]. The clinical features of migraine attacks are prevalently unilateral, throbbing head pain, with sensitivity to visual, auditory, and other input (e.g., olfactory) [2,11,12]. Prodromic and postdromic symptoms such as fatigue, food craving, irritability, yawning, and reduced concentration can impair the so-called interictal period, between two pain phases [13]. Furthermore, a variable percentage of migraine patients have their attacks associated with transitory neurological symptoms (prevalently visual), collectively named migraine aura [14,15].

In other words, recognizing the multifaced aspects of the clinical presentation of migraine can be challenging. Pain might not be the pivotal clinical symptom and can evolve with time, sometimes paradoxically less troublesome than other features. The background literature clearly imposes to go beyond the headache, and considering migraine exclusively as a pain-processing disorder is extremely simplistic misleading, affecting its management [16].

The comorbidity between migraine and psychiatric disorders has been extensively explored in the literature [17,18,19,20,21,22,23,24]. The strongest association was described with depression and anxiety, which seem to have a bidirectional relationship with migraine. Subjects with a combination of major depression and anxiety disorders are more likely to have migraine compared with those with depression or anxiety only and without both [2,22].

Fatigue has also been recognized as a dominant feature of migraine [25]. It is estimated that approximately 60% of migraineurs report pathologic fatigue [26]. Furthermore, fatigue seems to be correlated with symptoms of depression and headache intensity in migraineurs [26].

Depression, anxiety, and fatigue are pivotal elements of migraine-related disability and disease progression, regarded as risk factors for transforming EM into CM [3,27].

Prophylactic treatments for migraine can be also effective on psychiatric comorbidities (e.g., antidepressants) that are considered when choosing the proper drug for headache frequency control [27,28]. The role of novel targeted anti-migraine drugs in this fearful triad (depression, anxiety, and fatigue) is still unclear and unpredictable based on the putative pathophysiological mechanism.

We are still largely ignorant about migraine causative molecules and signaling pathways, although findings on both clinical and basic neuroscience have identified the central role of the activation of the trigeminovascular system and the antidromic release of molecules, such as the calcitonin gene-related peptide (CGRP) [28,29,30]. CGRP is released by small unmyelinated sensory (C-)fibers whose hyperactivation could result in what is called neurogenic inflammation [28].

Other neuropeptides and molecules contribute to the migraine pathophysiology and can be found in the innervation of cranial vessels (intra- and extracerebral), such as vasoactive intestinal polypeptide (VIP), nitric oxide (NO), adenylate cyclase-activating peptides (PACAP), amylin, and ions, although their role needs to be further verified [31]. Clinical studies have supported the importance of PACAP the pathophysiology of migraine but the clinical trials are still inconclusive with both a receptor monoclonal antibody (AMG 301) that showed no effects in preventing migraine, and a PACAP ligand monoclonal antibody (Lu AG09222) that showed promising results on headache frequency bit is still in Phase 2 clinical trial [32].

Compared to other neuropeptides, CGRP levels were consistently elevated in external jugular vein blood samples during the pain phase of migraine [33] and 5-hydroxytryptamine (5-HT, serotonin) receptors (5-HT1B and 5-HT1D) agonists (e.g., triptans) are effective suppressors of CGRP release and significantly abort migraine attacks [34].

Serotonin reuptake is also one of the main targets of the pharmacological treatment of psychiatric syndromes [35]. CGRP and serotonin signaling are tightly intertwined in the migraine pathophysiology, still little is known about the effects of CGRP blockage and migraine comorbidities [36,37] that could shed light on the common pathophysiology of the diseases and translate the results of CGRP-interference on mood disorders.

According to most clinical trials and real-world studies, the efficacy of novel treatments targeting the CGRP is not significantly different between EM and CM [38,39,40], including patients experiencing previous preventive treatment failures that were considered most difficult to treat [41,42]. However, real-world trials showed non-responders more numerous in CM [43,44], probably because CM patients with continuous pain and medication overuse headache (MOH) have been excluded from most clinical trials [45]. It has been reported in real-world studies that high frequency is not a negative predictor, but daily headache could prognosticate the failure of novel treatments [27]. In this context, we evaluated comorbid symptoms of anxiety, depression, and fatigue at baseline and 3 months after starting a treatment with monoclonal antibodies (mAbs) acting on the CGRP (Fremanezumab and Galcanezumab) or its receptor (CGRPR) (Erenumab) in a cohort of 77 subjects with a diagnosis of migraine resistant to traditional drug prophylaxis either high-frequency EM (HFEM > 8 headache days/month) or CM with or without MOH. Additionally, we decided to consider anxiety–depressive symptoms and fatigue as potential prognostic factors for drug efficacy. Hence, we sorted the patients into two groups according to whether the treatment was clinically effective or not, to observe if any differences in these symptoms were present at baseline.

## 2. Methods

### 2.1. Study Design

The study was an open-label prospective study evaluating symptoms of anxiety, depression, and fatigue in 77 HFEM and CM patients resistant to traditional prophylaxis and, therefore, undergoing treatment with anti-CGRP/R mAbs (Erenumab 70 or 140 mg/month, Galcanezumab 120 mg/month, and Fremanezumab 225 mg/month, administered by subcutaneous injection once a month). The study consisted of a preliminary evaluation (V0) to verify the eligibility of the patient, a baseline assessment (V1) with the first administration of anti-CGRP/R mAbs, and a 3-month follow-up visit (V2). The selected study period was conceived to observe the rapid effects of the novel molecules on migraine comorbidities. No acute medication was used during the visits. Information on acute medication used by the participants during the study period was recorded.

### 2.2. Study Participants

We enrolled adult outpatients, consecutively enrolled from November 2021 to December 2022 in the Neurology Unit, Center for Diagnosis and Treatment of Headaches and Craniofacial Pain at the University of Pisa, suffering from HFEM or CM according to the International Classification of Headache Disorders—3rd edition (ICHD-3) [46], and resistant to the common prophylaxis therapies according to the European Headache Federation (EHF) Consensus [47].

All patients received and failed at least three preventive medication classes (beta-blockers, calcium-channel blockers, anticonvulsants, antidepressants, onabotulinumtoxinA) due to a lack of efficacy or intolerable side effects.

Patients were discontinued from other preventive treatments at least 3 months before the baseline or were treated with stable oral migraine prophylaxis (defined as stable dosage of the medication for at least 6 months before the inclusion visit and for the duration of the study). No specific medication for psychiatric comorbidities was allowed or used by the participants. The exclusion criteria were: (1) age under 18; (2) diagnosis of schizophrenia, chronic psychosis, acute psychosis; (3) diagnosis of somatic and related symptom disorders; (4) diagnosis of ongoing substance use disorders; (5) patients with impaired speech; (6) patients with mental retardation; (7) diagnosis of other neurological diseases; (8) patients unable to provide valid written informed consent; (9) pregnant or breastfeeding patients; (10) patients with a desire to become pregnant during the study period.

This study was performed in accordance with the Declaration of Helsinki, and it was approved by the local ethics committee (Comitato Etico Area Vasta Nord Ovest—Sezione Autonoma del Comitato Etico Regionale per la Sperimentazione Clinica—Via Roma 67, 56126, Pisa, Italy) with approval code ID-14.518. All subjects involved provided written, informed consent before their inclusion.

### 2.3. Clinical Assessment

The study visits were performed before (V1) and after (V2) the administration of the treatment. Clinical characteristics of migraine were collected through an interview and based on patients’ self-reported diaries. Moreover, the following questionaries were administered at each visit: Migraine Disability Assessment (MIDAS) [48,49] for the evaluation of migraine-related disability, Fatigue Severity Scale (FSS) [50,51] to assess migraine-associated fatigue, the Generalized Anxiety disorder (GAD-7) [52], and Patient Health Questionnaire (PHQ-9) [53], to monitor anxiety and depressive symptoms, and Allodynia Symptoms Checklist 12 (ASC-12) to report ictal allodynia [54]. The questionnaires were administered by trained neurologists and neurology residents of a tertiary care outpatient clinic, specialized in the diagnosis and treatment of headaches and craniofacial pain. The neurologists administered the questionnaires properly translated into Italian.

### 2.4. Statistical Analysis

All demographic and clinical data were presented for continuous variables in terms of medians and interquartile ranges.

The quantitative variables of the sample, evaluated with the Shapiro–Wilk test, did not have a normal distribution. For this reason, to test the possible differences before and after the treatment with anti-CGRP/R mAbs, the Wilcoxon rank test was used, whereas to compare the two subgroups identified (responders and not-responders to anti-CGRP/R mAbs) the Mann–Whitney test was used.

Categorical variables were expressed as percentages and the comparison was performed by the Chi-square test with continuity correction (Yates test). Binary logistic regression analysis was performed to predict the likelihood of the patients responding to the anti-CGRP/R mAbs, according to the measured variables.

The differences were considered statistically significant for values of probability *p* < 0.05 (two tails). IBM Corp. Released 2016. IBM SPSS Statistics for Windows, Version 24.0. Armonk, NY, USA: IBM Corp was used for statistical analyses.

## 3. Results

The study population was composed of 77 patients, of which 59 (77%) were females, with a median age of 49.0 years old (IQR 15.0).

All patients were diagnosed with migraine without aura; 4 subjects (5%) had a concomitant diagnosis of migraine with aura and, according to the ICHD-3, were coded as both migraine with aura and migraine without aura. At baseline, 16 patients (21%) self-reported a frequency compatible with HFEM, and 61 (79%) with CM of which 43 were also diagnosed with medication overuse headache (MOH). MOH diagnosis was not reassessed at V2, as the duration of the symptoms required for the diagnosis was longer than the study period. All patients were resistant to traditional drug prophylaxis.

All patients were treated with mAbs therapy: 44 patients (57%) received monoclonal antibodies acting on the CGRP (12 were treated with Fremanezumab and 32 with Galganezumab) and 33 patients (43%) received monoclonal antibodies acting on the CGRPR (Erenumab).

For all patients, we evaluated comorbidity symptoms of anxiety, depression, fatigue, and allodynia at baseline and 3 months after starting the treatment. All patients completed the study. No adverse events, tolerability, or safety issues were reported.

The overall analysis showed that the treatments were highly effective in reducing migraine frequency which dropped from a median of 23 days/month to 6 days/month (*p* < 0.001). The same highly significant impact was registered for migraine-related disability, anxiety and depressive symptoms, allodynia, and fatigue, as summarized in Table 1.

We decided to run a subgroup analysis dividing the population at baseline into two sets (responders and non-responders), based on the clinical effectiveness of the drugs measured at V2. According to the EHF treatment guidelines, non-responders were defined as subjects that did not have a reduction of at least 50% in the frequency of migraine after the administration of drugs for at least three months. We identified 21 non-responders (27%) and 56 responders (73%).

The two groups were homogeneous, without distinctive features among the analyzed variables. The burden of disease, the distribution of mild psychiatric symptoms (anxiety and depression), fatigue, and ictal allodynia did not show significant differences (Table 2). The same result was obtained considering crude scores of the tests or clustering raw data according to validated clinical significance as absent, mild, moderate, and severe (from 0 to 3 in crescent order).

The responders’ group exhibited a reduction of disease severity, non-exclusively in terms of headache frequency but also associated disability, allodynia (*p* < 0.001), and psychiatric comorbidities (Table 3).

Non-responders showed a significant reduction of the headache frequency although below the 50% threshold (*p* = 0.003), and reduced migraine-related disability (*p* < 0.001). No significant improvement was registered for ictal allodynia, depressive symptoms, and fatigue. Although the raw scores for anxiety did not significantly change for this group, the clinical classification varied from a median value of moderate to mild symptoms during the 3 months of observation (*p* = 0.034).

However, the homogeneity of the groups did not allow for the prediction of treatment outcomes based on the investigated characteristics (Table 4).

## 4. Discussion

The overall analysis of the study indicated that the efficacy of mAbs targeting the CGRP in a real-life study for migraine prevention was combined with a significant improvement in psychiatric symptoms. Of note, the studied population was exclusively composed of HFEM or CM, resistant to the common drug prophylaxis prophylaxes (with or without the concomitant diagnosis of MOH).

A subgroup analysis was undertaken in patients who experienced ≥ 50% reductions in headache frequency (defined as responders) [41] after anti-CGRP/R mAbs treatment to detect clinical predictive factors. The relationship between anxiety, depression, fatigue, and migraine has been reported in several investigations. However, it is not clear whether the CGRP neurotransmission pathway may be directly involved in tuning affective symptoms [28]. The pain reduction due to anti-CGRP treatment could induce modification of “pain matrix” activity in the central nervous system and indirectly restore a neurotransmitter imbalance (e.g., serotonin and dopamine), pivotal for mood disorders [17]. CGRP could be related to chronic pain sensitization and cortical hyperexcitability, combined with other factors such as oxidation/reduction (redox) state [18,55,56]. In this framework, a chronic pain condition reporting several therapeutical failures represents, per se, a risk factor for psychiatric disease onset mining personal resilience. Furthermore, a link between migraine and psychiatric symptoms has been associated to genetic [57] and neurotransmitter modifications [58]. Altered endocannabinoid levels and decreased cerebrospinal fluid levels of GABA in CM patients with comorbid depression [59,60] may be part of the common pathophysiological signaling underlying affective comorbidity within migraine.

CGRP receptors, on the other hand, are usually relegated to the trigeminovascular system for the description of their causative role in migraine [28,29]. However, CGRP signaling is also found in the superior and inferior colliculi (phonophobia and photophobia), stria terminalis (anxiety-like behavior), hypothalamus (appetite regulation), thalamus (allodynia), amygdala, cerebellum, and neocortex (anxiety and depression), pointing towards multiple anatomic and functional interactions between migraine and its comorbidities [61,62,63,64].

Headache disorders, according to the global burden of disease (GBD) are the third most prevalent cause of global disability, expressed as years lived with disability (YLDs), just below depressive disorders if considering all genders and ages. The selection of young adults (age 15–49) of both genders makes headaches the most impacting condition in this stage of life, overcoming mood disorders [1]. We did not assess the pediatric (children and adolescents) population of migraineurs. The opportunity to treat these young patients with anti-CGRP/R drugs is one of the unsolved questions for the novel treatments [65]. One of the main issues is the physiological role of CGRP in bone formation. However, the rapid efficacy of these molecules, corroborated by our data, could allow treatment for short-term periods (e.g., 3 months), reducing the risk of metabolic interference.

Considering that the anti-CGRP/R antibodies are ineffective in a significant proportion of patients and are costly, it would be useful for the headache specialist, the patient, and public healthcare to be able to predict who will probably benefit from the molecule. This has always been a problem in migraine therapy and a tailored approach was based on comorbid conditions or contraindications more than on presumed efficacy. Predictors of effect have been retrospectively identified in clinical trials or real-world studies, but their clinical usefulness is scarce or there is none on the single patient, considering the measured predictive values [37,39,43,66].

Previous failures of preventive treatments, according to most trials, do not represent a negative predictor of anti-CGRP/R antibodies success, and our study confirms the evidence [41,65]. It should be noted that an inverse relationship between the responder rate and number of prior treatments was reported in other real-world studies [43,67,68]. It should be mentioned that Erenumab was tested in the LIBERTY trial with a lower responder rate (30% compared to 50%) after two to four previous treatment failures [69,70].

CM patients with daily headache, classified in the ICHD-3 as A1.3.2, may be poor responders to anti-CGRP/R antibodies as published after a compassionate use of Erenumab [43]; very low responder rates were found in this cohort of patients. These data are paralleled by higher rates of response in CM patients with pain-free periods (A1.3.1). Other real-world studies have found negative response rates in such patients [66], even when switching the molecules that were used [71]. We did not check for the A1.3.2 vs. A1.3.1 subgroups of CM in our cohort; however, the headache frequency was not a predictor of poor outcome in our population. More studies need to be conducted to understand if CM patients with continuous pain or MOH are less prone to responding to neuromodulation of CGRP signaling [72]. Psychiatric comorbidities were considered possible culprits and concomitant depression was more frequent in non-responders than in super-responders [44], but our data are not in line with this hypothesis. The baseline migraine frequency and interictal allodynia were also found as poor outcome predictors, however, our data do not support these results [66,73].

Other features that were identified as good predictors were not found in our real-world study, in particular headache unilaterality, less severe disability at baseline, good response to triptans, typical migraine features and vomiting, and young age [44,65,66].

Interestingly the responders and non-responders in this study had a similar age, burden of disease, and associated comorbidities. This evidence, on the one hand, further supports the homogeneity of the selected population and reduces the prognostic values of these characteristics but, on the other hand, suggest a common pathogenetic pathway between migraine and psychiatric comorbidities. This study provided additional evidence about mAbs anti-CGRP/R efficacy. In particular, the treatment consistently reduced allodynia, from moderate to absent in responders, whilst the non-responder group was unchanged. This observation may imply a modification of the central circuitry for the conscious perception of pain and cortical excitability [55,74,75,76], probably through the reduction of peripheral sensitization in responders [56,77,78]. Indeed, it was recently described that distinct thalamocortical circuits underlie allodynia induced by depression-like state rather than tissue damage [74]. Our group and others described a reduction in the allodynic symptoms in CM using onabotulinumtoxinA. The mechanisms of these preventive treatments could be convergent in reducing the peripheral and central excitability of the trigeminovascular/thalamic relays [78,79].

Our study has some caveats. First, a precise psychiatric diagnosis was not extensively investigated. Then, the sample number was relatively small, although quite homogenous, including only individuals reporting disabling migraine with a serious negative impact on daily life. The main strength of the study is the evaluation of the impact of monoclonal antibodies targeting the CGRP/R on comorbid symptoms of depression, anxiety, and fatigue in migraine patients who are resistant to conventional prophylaxis. This was an open-label prospective study, with a valuable contribution to the real-world data on the early efficacy on a specific and homogeneous population of patients.

The subgroup analysis responders versus non-responders offered insights into potential clinical predictive factors. There was a remarkable reduction of MIDAS in responders (more than 10-fold from V1 values, see Table 3) with the median score classified as mild disability at V2. The non-responders also experienced a significant reduction in the MIDAS score (2.6-fold), being still over the threshold of severe disability (≥21). Both groups of patients had baseline scores of disabilities (median above 85), substantially out-of-scale compared to the typical migraine patients. The cutoff levels were indeed difficult to apply, and the perceived amelioration of the personal disability was still of great impact also in non-responders.

The opposite can be said for anxiety and depressive symptoms. The responders’ group showed, after three months of treatment, a significant reduction in the median values of the relative questionnaires; however, the scores of the symptoms can still be categorized as mild in GAD7 while passing from mild to absent for PHQ9 [52,53]. Notably, the anxiety symptoms for non-responders improved from moderate to mild during the period of the study (*p* = 0.034), even if crude scores showed a non-significant reduction (*p* = 0.130). This observation needs to be confirmed and considered with caution, as the responder and non-responder groups did not significantly differ for both clinical grading and scores of GAD7. Longer follow-up might detect if migraine modifications affect the onset or reduction of psychiatric symptoms or vice versa. However, the strict relationship between migraine, anxiety, and depressive disorders is confirmed in our study [19,80].

Fatigue that can be also subtended by thalamocortical dysfunctional mechanisms in migraine, was notably reduced only in the responders’ group with a decrease of 33% from the baseline values [18,81,82]. The FSS scale has no clinically validated cutoffs. As for allodynia, it seems that the responders’ group had a highly significant reduction compared to non-responders. The FSS score reduction in migraine patients responding to anti-CGRP antibodies could be due to the improvement of the dysfunctional migraine-related mechanisms but could also suggest the potential role of CGRP neurotransmission in fatigue and dysfunctional pain-perception syndromes if these findings are replicated in further studies. This study increases the understanding of the potential advantages of the anti-CGRP/R mAbs in the treatment of both migraine and related affective, fatigue, and hyperalgesic comorbidities.

## 5. Conclusions

The role of the novel prophylactic agents for migraine targeting the CGRP system is not limited to the improvement of disease severity but also affects anxiety, depressive symptoms, and fatigue. Among those conditions, allodynia and fatigue seem to be responsive to these treatments in those patients who experienced the highest clinical impact. The clinical benefits are remarkably precocious, and the non-responders still manifest high clinical impacts on their quality of life. There are still open questions about the therapeutical opportunities of these new molecules, and we assessed a relatively unexplored field of study. Our results are hastening the interlocked bidirectional link between affective disorders and migraine. The future direction of CGRP studies should try to unravel a common pathophysiological mechanism for interictal and ictal manifestations of migraine and its comorbidities. These common features could speculatively open a new field of investigation for the CGRP neurotransmission that goes beyond pain modulation and directly affects the resilience of the CNS.

## Figures and Tables

**Table 1 biomedicines-12-00677-t001:** Clinical assessment at baseline and after 3 months.

	V1 Raw Data		Clinical Score (75–25 Percentile)	V2 Raw Data		Clinical Score (75–25 Percentile)	
	Median	75–25 Percentiles		Median	75–25 Percentiles		*p* ^1^
**Age (years)**	47.00	57–41					
**Headaches frequency** **(days/month)**	23.00	30–15		6.00	13–4		**<0.001 ***
**MIDAS**	93.00	133–69		24.50	49–4		**<0.001 ***
**ASC-12**	6.00	10–4	Moderate (3.00–1.00)	3.00	7–0	Mild (2.00–0.00)	**<0.001 *#§**
**GAD7**	9.00	13–6	Mild (2.00–1.00)	6.50	9–4	Mild (1.75–0.00)	**<0.001 *#§**
**PHQ9**	8.00	13–6	Mild (2.00–1.00)	6.00	9.00–3.00	Mild (1.00–0.00)	**<0.001 *#§**
**FSS**	46.00	57–34		36.00	44–20		**<0.001 ***

^1^ Test Wilcoxon; MIDAS: Migraine Disability Assessment Scale; FSS: Fatigue Severity Scale; ASC-12: Allodynia Symptom Checklist 12; GAD7: Generalized Anxiety Disorder 7; PHQ9: Patient Health Questionnaire 9; Clinical score:. Severe = 3; Moderate = 2; Mild = 1; Absent = 0; # raw data; § clinical score (* *p* < 0.001).

**Table 2 biomedicines-12-00677-t002:** Clinical features of responders and non-responders at baseline.

	Responder (% of All Patients)	Non-Responder (% of All Patients)	
	N (%)		*p* ^1^
**EM**	13 (17%)	3 (4%)	0.586
**CM**	43 (56%)	18 (23%)	
**Patients with MOH**	34 (44%)	9 (12%)	0.981
**Patientes without MOH**	22 (28%)	12 (16%)	
	**Median (75–25 percentiles)**		***p*** ^**2**^
**Age (years)**	50.00 (57–44)	41.00 (49–36)	0.082
**Headaches frequency** **(days/month)**	22.00 (30–14)	30.00 (30–15)	0.237
**MIDAS**	90.00 (129–67)	99.00 (152–78)	0.171
**ASC-12** **(raw data)**	6.00 (10–4)	5.00 (12–3)	0.812
**ASC-12** **(clinical score)**	Moderate (3–1)	Mild (3–1)	0.628
**GAD7** **(raw data)**	8.00 (13–6)	10.00 (14–6)	0.491
**GAD7** **(clinical score)**	Mild (2–1)	Moderate (3–1)	0.250
**PHQ9** **(raw data)**	7.50 (13–5)	10.00 (14–6)	0.132
**PHQ9** **(clinical score)**	Mild (2–1)	Moderate-Mild (2–1)	0.259
**FSS**	45.00 (57–33)	54.00 (59–39)	0.151

^1^ Chi-Squared test; ^2^ U Mann–Whitney test; MIDAS: Migraine Disability Assessment Scale; FSS: Fatigue Severity Scale; ASC-12: Allodynia Symptom Checklist 12; GAD7: Generalized Anxiety Disorder 7; PHQ9: Patient Health Questionnaire 9; EM: episodic migraine; CM: chronic migraine; MOH: medication overuse headache; Clinical score:. Severe = 3; Moderate = 2; Mild = 1; Moderate-Mild = 1.5; Absent = 0.

**Table 3 biomedicines-12-00677-t003:** Differences between parameters at baseline and after three months in responders and non-responders.

	Responder Median (75–25 Percentiles)			Non-Responder Median (75–25 Percentiles)		
	V1	V2	*p* ^1^	V1	V2	*p* ^1^
**Headaches frequency** **(days/month)**	22.00 (30–14)	5.00 (6–3)	**<0.001 *****	30.00 (30–15)	20.00 (30–12)	**0.003 ****
**MIDAS**	96.00 (135–70)	8.00 (39–2)	**<0.001 *****	99.00 (15278)	38.00 (60–26)	**<0.001 *****
**ASC-12** **(raw data)**	6.00 (9–3)	1.50 (5–0)	**<0.001 *****	5.00 (12–2)	6.00 (9–1)	0.241
**ASC-12** **(clinical score)**	Moderate (3–1)	Absent (1–0)	**<0.001 *****	Mild (3–0)	Mild(3–0)	0.476
**GAD7** **(raw data)**	8.00 (12–6)	6.00 (7–3)	**0.001 ****	10.00 (13–5)	7.00 (10–6)	0.130
**GAD7** **(clinical score)**	Mild (2–1)	Mild (1–0)	**0.002 ****	Moderate (2–1)	Mild (2–1)	**0.034 ***
**PHQ9** **(raw data)**	7.00 (12–5)	4.00 (7–2)	**<0.001 *****	10.00 (14–6)	8.00 (13–6)	0.184
**PHQ9** **(clinical score)**	Mild (2–1)	Absent (1–0)	**<0.001 *****	Mild-Moderate (2–1)	Mild (2–1)	0.414
**FSS**	43.50 (57–26)	29.00 (40–17)	**0.001 ****	54.00 (61–44)	42.00 (55–39)	0.109

^1^ Wilcoxon test; MIDAS: Migraine Disability Assessment Scale; FSS: Fatigue Severity Scale; ASC-12: Allodynia Symptom Checklist 12; GAD7: Generalized Anxiety Disorder 7; PHQ9: Patient Health Questionnaire 9; Clinical score:. Severe = 3; Moderate = 2; Mild = 1; Moderate-Mild = 1.5; Absent = 0 (* *p* < 0.5; ** *p* < 0.01; *** *p* < 0.001).

**Table 4 biomedicines-12-00677-t004:** Logistic regression predicting the likelihood of responding to therapy with anti-CGRP/R mAbs at baseline (responders vs non-responders).

	*p*	OR	95% CI	
			Lower	Upper
**Sex**	0.223	0.406	0.095	1.731
**Age**	0.225	1.030	0.982	1.081
**ASC12**	0.619	1.036	0.901	1.191
**FSS**	0.374	0.978	0.932	1.027
**GAD7**	0.485	1.070	0.885	1.292
**PHQ9**	0.158	0.886	0.748	1.048

FSS: Fatigue Severity Scale; ASC-12: Allodynia Symptom Checklist 12; GAD7: Generalized Anxiety Disorder 7; PHQ9: Patient Health Questionnaire 9; OR: odds ratio.

## Data Availability

The data presented in this study are available on request.

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
