# Peer review of "Beyond Pain Relief: Unveiling the Multifaceted Impact of Anti-CGRP/R mAbs on Comorbid Symptoms in Resistant Migraine Patients"

_biomedicines, 2024, doi:10.3390/biomedicines12030677_

Round 1

Reviewer 1 Report

Comments and Suggestions for Authors

The study by Alessandra Della Vecchi is interesting, and I only have a few comments/questions:

Line 123 “ All patients were diagnosed with migraine without aura, 4 subjects (5%) had a con-comitant diagnosis of migraine with aura. How can 5% have both?

Line 125, this seems to distort the data “with CM of which 43 were also diagnosed with medication overuse headache (MOH).” What happened with the medication use throughout your study?

Line 130, Name the antibody in “received monoclonal antibodies acting on the CGRPR”

The percent in table 2 is difficult to follow, this is a percent of what? I.e. what is 100%

In table 2, there are 34 without MOH, I guess these include the EM, without MOH, if not the numbers add up, but this is confusing, as it is jointly written under CM?

Line 164, “Non-responders showed a significant reduction of the headache frequency although 163 below the 50% threshold” I understand that this was your definition, but are they then really non-responders, or weak-responders?

Did you evaluate headache intensity or only frequency? There might be a difference in intensity in your “non-responders” ?

Author Response

The study by Alessandra Della Vecchi is interesting, and I only have a few comments/questions:

Line 123 “ All patients were diagnosed with migraine without aura, 4 subjects (5%) had a concomitant diagnosis of migraine with aura. How can 5% have both?

Thank you for the comment. The ICHD-3 classification we based our study on, clarifies that “Many patients who have migraine attacks with aura also have attacks without aura; they should be coded as both 1.2 Migraine with aura and 1.1 Migraine without aura.”. According to ICHD-3 these patients receive both diagnoses.

Line 125, this seems to distort the data “with CM of which 43 were also diagnosed with medication overuse headache (MOH).” What happened with the medication use throughout your study?

We considered the observation period of three months not sufficient to reassess the MOH diagnosis that lasts longer than 3 months (criterion B). We added a line to specify it within the manuscript.

Line 130, Name the antibody in “received monoclonal antibodies acting on the CGRPR”

We added Erenumab as suggested.

The percent in table 2 is difficult to follow, this is a percent of what? I.e. what is 100%

The percentage refers to the totality of the patients, we specified it in the titles row to fulfill your suggestion.

In table 2, there are 34 without MOH, I guess these include the EM, without MOH, if not the numbers add up, but this is confusing, as it is jointly written under CM?

We separated the MOH and CM rows with marked lines and specified that these are patients with MOH (CM by definition) or without MOH (con be either CM or EM).

Line 164, “Non-responders showed a significant reduction of the headache frequency although 163 below the 50% threshold” I understand that this was your definition, but are they then really non-responders, or weak-responders?

We agree with your comment, a clinical benefit is registered also in the non-responder group. However, we used the EHF treatment guidelines to set the groups and the response threshold.

Did you evaluate headache intensity or only frequency? There might be a difference in intensity in your “non-responders” ?

The intensity was assessed in the MIDAS but not evaluated as a single variable. The MIDAS showed a significant reduction also in non-responders.

Reviewer 2 Report

Comments and Suggestions for Authors

The manuscript presented titled: “ Early Improvement of Affective, Fatigue and Allodynic Symptoms in a Cohort of Resistant Migraineurs Treated with Anti-CGRP/R Antibodies”

 The authors have conducted open-label prospective study to determine in resistant to traditional prophylaxis for migraine treatment the effecacy of anti-CGRP/R mAbs. The most significant result of the study conducted in a good numeber of patients, is the absence of adverse side effects of its use, even if it was evaluated after 3 months (short term, since there are study at 9 months). Can anti-CGRP/R mAbs be used in adolescents and children? The efficacy you report afetr 3 months can candiates it also in pediatric population. Discuss this point.

Minor comments:

Uniforms the manuscript with the same characters and letter size. See line 92

Explain better the p value in the tables equals to 0.000***

1. The paper is well-referenced, but incorporating studies in the discussion section on pediatric population must be.

This is a short list of previous reports on this topic:

- Schoenen, J., Van Dycke, A., Versijpt, J. et al. Ten open questions in migraine prophylaxis with monoclonal antibodies blocking the calcitonin-gene related peptide pathway: a narrative review. J Headache Pain 24, 99 (2023). https://doi.org/10.1186/s10194-023-01637-7

- Krymchantowski AV, Jevoux C, Krymchantowski AG, Silva-Néto RP. Monoclonal antibodies for chronic migraine and medication overuse headache: A real-world study. Front Neurol. 2023 Mar 3;14:1129439. doi: 10.3389/fneur.2023.1129439. PMID: 36937507; PMCID: PMC10022428.

Author Response

The manuscript presented titled: “ Early Improvement of Affective, Fatigue and Allodynic Symptoms in a Cohort of Resistant Migraineurs Treated with Anti-CGRP/R Antibodies”

 The authors have conducted open-label prospective study to determine in resistant to traditional prophylaxis for migraine treatment the effecacy of anti-CGRP/R mAbs. The most significant result of the study conducted in a good numeber of patients, is the absence of adverse side effects of its use, even if it was evaluated after 3 months (short term, since there are study at 9 months). Can anti-CGRP/R mAbs be used in adolescents and children? The efficacy you report afetr 3 months can candiates it also in pediatric population. Discuss this point.

We thank the reviewer for opening the discussion on the pediatric population, we covered this topic in the discussion. We better specified in the study design that the 3 month-period was for early efficacy detection.

Minor comments:

Uniforms the manuscript with the same characters and letter size. See line 92

We uniformed the paragraphs' styles

Explain better the p value in the tables equals to 0.000***

To avoid misinterpretation, we corrected the value as <0.001 substituting 0.000 in all tables.

  1. The paper is well-referenced, but incorporating studies in the discussion section on pediatric population must be.

This is a short list of previous reports on this topic:

- Schoenen, J., Van Dycke, A., Versijpt, J. et al. Ten open questions in migraine prophylaxis with monoclonal antibodies blocking the calcitonin-gene related peptide pathway: a narrative review. J Headache Pain 24, 99 (2023). https://doi.org/10.1186/s10194-023-01637-7

- Krymchantowski AV, Jevoux C, Krymchantowski AG, Silva-Néto RP. Monoclonal antibodies for chronic migraine and medication overuse headache: A real-world study. Front Neurol. 2023 Mar 3;14:1129439. doi: 10.3389/fneur.2023.1129439. PMID: 36937507; PMCID: PMC10022428.

We added the requested references

Reviewer 3 Report

Comments and Suggestions for Authors

This is a very interesting study aiming to evaluate the effects of anti-CGRP mAbs on comorbid symptoms of depression anxiety and fatigue in migraine patients.

Overall, this study is a well-organized and conducted study with interesting results.

Some minor revisions are needed:

Line 39: …literature (citation is needed)

Line 44: …fatigue (citation is needed)

Line 190: citation is needed.

Lines 92-96 font is larger

Please report some information regarding each questionnaire used and cite original publications of them. In what language did these questionnaires were administered? Are all these questionnaires properly translated in this language?

Please report who (and of what specialty) administered these questionnaires  .

Please report if any acute medication was used (if this information is available) .

Furthermore please report if any other medication for the psychiatric comorbidities was allowed/used.

The decimal point at Table 2 at Age, MIDAS, Headache frequency etc are confusing.

Please cite SPSS properly (IBM Corp. Released 2016. IBM SPSS Statistics for Windows, Version 24.0. Armonk, NY: IBM Corp)

At the Discussion section please discuss any pathophysiological theories regarding the effects of anti-CGRP mAbs on psychiatric comorbidities of migraine patients. Comment on the potential role of the endocannabinoid and serum serotonin levels alteration or any other potential mechanisms.

You could consult the following publications:

Smitherman, T. A., Tietjen, G. E., Schuh, K., Skljarevski, V., Lipsius, S., D’Souza, D. N., & Pearlman, E. M. (2020). Efficacy of galcanezumab for migraine prevention in patients with a medical history of anxiety and/or depression: a post hoc analysis of the phase 3, randomized, double‐blind, placebo‐controlled REGAIN, and pooled EVOLVE‐1 and EVOLVE‐2 studies. Headache: The Journal of Head and Face Pain, 60(10), 2202-2219.

Vikelis, M., Dermitzakis, E. V., Xiromerisiou, G., Rallis, D., Soldatos, P., Litsardopoulos, P., ... & Argyriou, A. A. (2023). Effects of fremanezumab on psychiatric comorbidities in difficult-to-treat patients with chronic migraine: post hoc analysis of a prospective, multicenter, real-world Greek registry. Journal of Clinical Medicine12(13), 4526.

Author Response

This is a very interesting study aiming to evaluate the effects of anti-CGRP mAbs on comorbid symptoms of depression anxiety and fatigue in migraine patients.

Overall, this study is a well-organized and conducted study with interesting results.

Some minor revisions are needed:

Line 39: …literature (citation is needed)

Line 44: …fatigue (citation is needed)

Line 190: citation is needed.

Lines 92-96 font is larger

We checked and added the requested information or updated the font

Please report some information regarding each questionnaire used and cite original publications of them. In what language did these questionnaires were administered? Are all these questionnaires properly translated in this language?

Please report who (and of what specialty) administered these questionnaires  .

We reported this information in the Clinical Assessment.

Please report if any acute medication was used (if this information is available) .

No acute medication was used during the visits. Information on acute medication used by the participants during the study period was recorded (data not shown).

Furthermore please report if any other medication for the psychiatric comorbidities was allowed/used.

No specific medication for psychiatric comorbidities was allowed or used by the partici-pants

The decimal point at Table 2 at Age, MIDAS, Headache frequency etc are confusing.

To avoid confusion, we rounded the percentiles.

Please cite SPSS properly (IBM Corp. Released 2016. IBM SPSS Statistics for Windows, Version 24.0. Armonk, NY: IBM Corp)

SPSS has been cited properly

At the Discussion section please discuss any pathophysiological theories regarding the effects of anti-CGRP mAbs on psychiatric comorbidities of migraine patients. Comment on the potential role of the endocannabinoid and serum serotonin levels alteration or any other potential mechanisms.

We widened the discussion, including the suggested theories

You could consult the following publications:

Smitherman, T. A., Tietjen, G. E., Schuh, K., Skljarevski, V., Lipsius, S., D’Souza, D. N., & Pearlman, E. M. (2020). Efficacy of galcanezumab for migraine prevention in patients with a medical history of anxiety and/or depression: a post hoc analysis of the phase 3, randomized, double‐blind, placebo‐controlled REGAIN, and pooled EVOLVE‐1 and EVOLVE‐2 studies. Headache: The Journal of Head and Face Pain, 60(10), 2202-2219.

Vikelis, M., Dermitzakis, E. V., Xiromerisiou, G., Rallis, D., Soldatos, P., Litsardopoulos, P., ... & Argyriou, A. A. (2023). Effects of fremanezumab on psychiatric comorbidities in difficult-to-treat patients with chronic migraine: post hoc analysis of a prospective, multicenter, real-world Greek registry. Journal of Clinical Medicine12(13), 4526.

These citations have been included

Reviewer 4 Report

Comments and Suggestions for Authors

Dear authors,

Thank you for submitting your interesting work about improvement after the use of mAbs in migraine patients. My comments:

The title would be more precise if you use "Treatment-resistant Migraineurs"...

I can’t understand who many of the 77 patients got every mAbs (who many Frema, galca or erenumab). Please be more specific. And were there differences in the results between anti-CGRP mAbs vs. anti-CGRP/R mAb (erenumab)?

L81: OnaA is not an oral preventive treatment.

L97:  In the “clinical Assessment” section it would be great if you could add the HIT-6 scale.

L123: do you mean: “73 patients were diagnosed with migraine without aura, 4 subjects (5%) had a …..” . Moreover, only 5% of the migraineurs had aura? it is a little low

In the Discussion session please comment more about the psychiatric comorbidities and compare if there are other related studies (for example Vikelis M. et al. J. Clin. Med., 2023) and separate from allodynia. For allodynia you could also compare with other studies (Lipton et al. Cephalalgia, 2021) or compare with the effect of OnaA on allodynia (Argyriou AA et al. Toxins, 2024).

Author Response

Dear authors,

Thank you for submitting your interesting work about improvement after the use of mAbs in migraine patients. My comments:

The title would be more precise if you use "Treatment-resistant Migraineurs"...

We thank you for the suggestion, since we want to stress the precise clinical diagnosis of “resistant migraine” according to the European Headache Federation (EHF) Consensus, we changed the title as: “…Patients with Resistant Migraine”

I can’t understand who many of the 77 patients got every mAbs (who many Frema, galca or erenumab). Please be more specific. And were there differences in the results between anti-CGRP mAbs vs. anti-CGRP/R mAb (erenumab)?

12 were treated with Fremanezumab, 32 with Galganezumab and 33 patients Erenumab. We specified it in the manuscript. We have not done a subgroup analysis stratifying for treatment.

L81: OnaA is not an oral preventive treatment.

Corrected

L97:  In the “clinical Assessment” section it would be great if you could add the HIT-6 scale.

We have not assessed the HIT-6 scale, we are afraid we cannot add it.

L123: do you mean: “73 patients were diagnosed with migraine without aura, 4 subjects (5%) had a …..” . Moreover, only 5% of the migraineurs had aura? it is a little low

We meant that 4 patients out of 77 (5%) have both diagnoses. The ICHD-3 classification we based our study on, clarifies that “Many patients who have migraine attacks with aura also have attacks without aura; they should be coded as both 1.2 Migraine with aura and 1.1 Migraine without aura.”

In the Discussion session please comment more about the psychiatric comorbidities and compare if there are other related studies (for example Vikelis M. et al. J. Clin. Med., 2023) and separate from allodynia. For allodynia you could also compare with other studies (Lipton et al. Cephalalgia, 2021) or compare with the effect of OnaA on allodynia (Argyriou AA et al. Toxins, 2024).

We widened the discussion about psychiatric comorbidities and added the convergent putative mechanism with BoNT/A

Round 2

Reviewer 4 Report

Comments and Suggestions for Authors

N/A

Author Response

N/A